# Effect of Slow-Release Urea on Yield and Quality of *Euryale ferox*

**DOI:** 10.3390/ijms252111737

**Published:** 2024-10-31

**Authors:** Peng Wu, Tian-Yu Wang, Yu-Hao Wang, Ai-Lian Liu, Shu-Ping Zhao, Kai Feng, Liang-Jun Li

**Affiliations:** 1School of Horticulture and Landscape Architecture, Yangzhou University, Wenhui East Road No. 48, Yangzhou 225009, China; wupeng@yzu.edu.cn (P.W.); 18852712804@163.com (T.-Y.W.); wangyuhaoyzu@163.com (Y.-H.W.); liuailianyz@163.com (A.-L.L.); zhaoshuping@yzu.edu.cn (S.-P.Z.); fengkai@yzu.edu.cn (K.F.); 2Joint International Research Laboratory of Agriculture and Agri-Product Safety of Ministry of Education of China, Yangzhou University, Yangzhou 225009, China

**Keywords:** *E. ferox*, SRU, SPAD value, chlorophyll content, starch content, soluble sugars, enzymes of starch synthesis, flavonoids

## Abstract

Slow-release urea, as an environmentally friendly fertiliser, can provide a continuous and uniform supply of nutrients needed by the crop, reduce the amount and frequency of fertiliser application, and promote the uptake and utilisation of nitrogen in crops. The production of *E. ferox* is often dominated by the application of quick-acting fertilisers, resulting in serious problems of over-fertilisation, inappropriate periods of fertilisation, eutrophication of soil and water due to fertilisation, and difficulties in applying fertilisers. Therefore, in this study, different amounts (CK, T1, T2, T3, T4, T5) of SRU (Slow-release Urea) were first applied, and T3 (18.8 kg·667 m^−2^) was found to significantly improve both yield and quality. Further, it was found that under different SRU (CK, S1, S2, S3, S4) application period treatments, application of 18.8 kg·667 m^−2^ at AFP20 (S2) period significantly increased the yield and quality of *E. ferox*. In the seed kernels of *E. ferox*, the total yield, soluble sugar content, total starch, and flavonoid content increased significantly by 10.35%, 36.40%, 5.91%, and 22.80%, respectively, compared with CK. In addition, the expression of key sugar transporter genes (*EfSWEETs*), flavonoid synthesis-related genes (*EfPAL*, *EfDFR*, etc.), and starch synthesis-related enzyme activities (SBE, SSS, GBSS) were significantly increased. By exploring the quantity of application and application period of SRU, this study was carried out to investigate the in-depth effect of SRU on the growth and development of *E. ferox* and to provide technical references for the increase in *E. ferox* yield, the improvement in *E. ferox* quality, and the simplification of fertiliser application.

## 1. Introduction

Nitrogen (N) is an essential element for all living organisms and an important factor in crop growth and productivity [1]. With the increasing demand for products, improving yield and quality has become an important issue. Therefore, farmers usually apply large amounts of nitrogen fertilizer to increase crop yields. However, most crops have low nitrogen-use efficiency, and more than half of the nitrogen fertilizer applied is lost before it is absorbed by the crop [2]. Therefore, increasing nitrogen use efficiency is an important measure to reduce nitrogen loss, increase crop yield, and improve crop quality in crop cultivation [3]. Slow-release urea (SRU) is a type of urea that has nitrogen, and the nitrogen is released in a slow manner with low nutrient loss, in sync with the plant’s nutrient needs [4].

The SRU can increase the quality and yield of crops [5]. SRU application can significantly increase rice yield and increase the number of panicles in rice [6]. Application of controlled-release nitrogen fertiliser significantly increased spring maize yields and increased nitrogen fertiliser utilisation by 16.3% [7]. Strawberry yields at the same level of nitrogen application were significantly increased by more than 10% on average under SRU treatment compared to regular urea [8]. Soybean applied with slow-release urea showed 21.7% higher yield and 12.0% higher nitrogen utilisation compared to regular urea [9].

On the other hand, SRU applications can significantly improve crop quality. After applying SRU, the total starch content of lotus (*Nelumbo nucifera*) rhizomes was significantly higher than that of urea treatment, which improved the quality of lotus flowers [10]. Rice had the highest content of straight-chain starch, branched-chain starch, and total starch after SRU application [11]. Moreover, the application of slow-release urea will affect the transport and accumulation of sugar in crops. Fertilisation treatments significantly increased pear (*Pyrus* spp.) yield and photosynthetic efficiency and significantly increased sucrose content in pear fruit [12]. Similarly, under the slow-release fertilizer treatment, the soluble sugar content of leeks increased by 16% compared with the control group [13].

Changes in starch accumulation or sugar translocation accumulation after the application of SRU are caused by changes in the expression of nitrogen-related genes. Among these, fertilization can affect the starch content and physicochemical properties of crops by regulating the expression of key genes of starch synthase [14]. Under SRU treatment, the changes in GBSS, SSS, and SBE activities in the early and middle stages of potato growth were consistent with the changes in their related genes, which play an important role in potato starch synthesis and accumulation [15]. The expression levels of starch-related genes *OsSBEII*, *OsGBSSI*, and *OsSBEI* were significantly decreased by SRU treatment, which reduced the ratio of straight-chain starch content to branched-chain starch content and improved the cooking quality of rice [16]. The conversion and accumulation of sugar, as a precursor to starch synthesis, is also affected by nitrogen [17,18]. The application of SRU increased the expression of *ZmSWEET11* and *ZmSWEET15* in maize leaves, thereby increasing maize grain fruit content and yield [19].

*Euryale ferox* Salisb. is an annual macrophyte aquatic herb that belongs to the genus *Euryale* from the family Nymphaeaceae [20]. *Euryale ferox* has a long history of cultivation in China as a kind of medicine and food plant [21]. *E. ferox* has high nutritional value. It is also used in Traditional Chinese Medicine based on the concept of qi (the “vital energy” or ‘life energy” of a body) as the component of many Chinese herbal medicines [22]. At the same time, *demand for Euryale ferox in the domestic and international markets* is rapidly increasing; the price is rising, and the development prospects are broad. SRU can synchronize nutrient release with nutrient absorption of *E. ferox*, simplify fertilization technology, and achieve one-time fertilization [23]. However, there is almost no research on fertilizer for *E. ferox*. In this study, the effect of SRU on the yield and quality of *E. ferox* was investigated by SRU application rates (CK, T1, T2, T3, T4, T5) and SRU application periods (CK, S1, S2, S3, S4). The results of this study provide a simpler and more effective method for further improving the yield and quality of *E. ferox* by fertilization.

## 2. Results

### 2.1. Effects of Different Quantities of SRU on the Chlorophyll Content of E. ferox

SRU application affected the chlorophyll content and SPAD values of *E. ferox* leaves. Chlorophyll content and SPAD values in *E. ferox* leaves increased and then decreased after SRU treatments at different quantities of application. When the SRU quantity of application was 18.8 kg·667 m^−2^ (T3), the chlorophyll content and SPAD value reached the maximum value, which was significantly higher than that of CK. The Chlorophyll content showed a sequence of T3 > T4 > T2 > T1 > T5 > CK, and the SPAD value showed a sequence of T3 > T4 > T5 > T1 > CK (Figure 1). Therefore, the maximum chlorophyll content and SPAD values of *E. ferox* leaves were reached when SRU was applied at 18.8 kg·667 m^−2^ (T3).

### 2.2. Effects of Different Quantities of SRU on the Yield of E. ferox

With the increase of SRU application, the net fruit weight, fruit diameter, single fruit grain weight, number of seed kernels, and seed kernel weight showed a trend of increasing and then decreasing and reached the maximum value of 761.52 g, 12.52 cm, 314.47 g, 147 kernels, and 61.40 g, respectively (Figure 2). They showed the sequence of T3 > T5 > T4 > T2 > T1 > CK, T3 > T5 > T4 > T2 > T1 > CK, T3 > T4 > T5 > T1 > T2 > CK, T3 > T4 > T5 > T1 > T2 > CK, and T3 > T4 > T5 > T1 > T2 > CK, respectively (Appendix A). In addition, the *E. ferox* yield showed an increasing and then decreasing trend with the increase of SRU application. When the fertiliser application is T3, *E. ferox* plant yield, as well as total yield, reached the maximum values of 7.80 kg and 32.10 kg, which increased by 17.09% and 16.45% compared with the CK, and showed a sequence of T3 > T5 > T2 > T1 > T4 > CK and T3 > T5 > T2 > T1 > T4 > CK, respectively (Appendix A). These results indicated that SRU applied at 18.8 kg·667 m^−2^ (T3) significantly promoted the growth and development of *E. ferox* fruits, thereby increasing the yield.

### 2.3. Effects of Different Quantities of SRU on the Quality of E. ferox

To elucidate whether SRU has any effect on *E. ferox* quality, we further investigated the effects of SRU treatment on soluble sugars, starch, and flavonoids in *E. ferox* seed kernels. The results showed that the soluble sugar content in the seed kernels of each fertilisation treatment showed a trend of increasing and then decreasing from DAF15 to DAF40. Particularly, the soluble sugar content of the T3 treatment reached a maximum value of 14.97 mg·g^−1^ at DAF20 and was significantly higher than the other treatments, with an increase of 40.13% compared with CK (Figure 3). Therefore, we further explored the relative expression of sugar transporter genes in *E. ferox* seed kernels at DAF20 in T3 treatment, using CK as a control. The results showed that the relative expression of *EfSWEET4.1.3*, *EfSWEET4.3*, *EfSWEET3.1*, *EfSWEET11.1*, *EfSWEET11.2*, *EfSWEET11.3* was significantly higher than that of CK, which was 14.2-fold, 3.7-fold, 3.5-fold, 16.2-fold, 8-fold and 2.9-fold, respectively (Figure 4). In conclusion, the soluble sugar content could be maximised by increasing the expression of sugar transport-related genes at SRU quantity of application of 18.8 kg·667 m^−2^ (T3).

Nitrogen treatment can change the starch content and fractions of crops [5]. In this study, changes in the starch content of *E. ferox* seed kernels were determined under different application conditions. The total starch content (Appendix A) and branched-chain starch content showed an increasing trend from DAF15 to DAF40, and all treatments reached the maximum value at DAF40. Meanwhile, the T3 treatment had the highest total starch and branched-chain starch content, which increased by 12.53% and 8.82% compared with the CK treatment. However, the content of straight-chain starch gradually decreased with the growth and development process of *E. ferox* (Appendix A), and there was no significant difference between the treatments. The results of the experiment showed that SRU application at 18.8 kg·667 m^−2^ (T3) significantly increased the total and branched-chain starch contents of *E. ferox* seed kernels. In addition, the relative expression of starch synthesis genes in *E. ferox* seed kernels was measured. The branched-chain starch and straight-chain starch synthesis genes *EfDBE*, *EfSBE*, *EfGBSS*, *EfSS2* reached their maximum values at DAF30, DAF25, DAF25, DAF25, and DAF25, which increased 1.8-fold, 7.29-fold, 8.7-fold, and 8.9-fold, compared with CK, respectively (Figure 4). Finally, the effects of SRU application on the activities of enzymes related to starch synthesis in *E. ferox* were determined. SBE and SSS enzyme activities showed a trend of a single peak curve, reaching the maximum at DAF25 and DAF30, respectively, which increased by 23.23% and 25.00% over CK (Appendix A). It indicates that T3 treatment not only significantly increased the relative expression of starch synthesis genes but also significantly increased SSS and SBE enzyme activities, thus promoting starch accumulation.

In addition, the flavonoid content was determined to investigate further the effect of SRU treatments on the quality of E. ferox seed kernels. The results showed that the content of flavonoid compounds in the seed kernels under each fertilisation treatment showed a gradually decreasing trend from DAF15 to DAF40 (Appendix A). The highest content of flavonoid compounds was observed under T3 treatment at DAF15, which increased by 50.13% compared to CK. Therefore, we selected DAF15 and determined the relative expression of flavonoid synthesis genes in *E. ferox* seed kernels from T3 and CK. The results showed that the relative expression of *EfPAL2*, *EfF3H3*, *EfDFR1*, *EfDFR2*, *EfLAR*, and *EfFLS* increased 2.6-fold, 0.6-fold, 0.4-fold, 0.5-fold, 1.8-fold, and 5.8-fold compared with CK (Figure 4). In summary, the optimal quantity of application of SRU was 18.8 kg·667 m^−2^. Apply less fertilizer and lower costs than traditional fertilization.

### 2.4. Effects of Different Periods of SRU on the Chlorophyll Content of E. ferox

In order to determine the scientific and reasonable fertiliser application period to achieve the effect of increasing yield and quality according to the nutrient requirements of crops at various stages of growth and development, combined with the nitrogen release characteristics of SRU [5]

. In this study, we further measured the yield, flavonoid, starch, and soluble sugar content of *E. ferox* after a one-time application of 18.8 kg·667 m^−2^ of SRU at different application periods in order to clarify the effect of SRU application period on the yield and quality of *E. ferox*.

The comparatively derived quantity of application of 18.8 kg·667 m^−2^ was used to treat *E. ferox* with SRU for different periods. The results showed that the chlorophyll content and SPAD values of *E. ferox* leaves showed a trend of increasing and then decreasing. The chlorophyll content and SPAD value reached the maximum value after S2 treatment, which increased by 16% and 12% compared with CK, respectively. The Chlorophyll content showed a sequence of S2 > S1 > S4 > S3 > CK, and the SPAD value showed a sequence of S2 > S4 > S1 > S3 > CK (Figure 5). The above findings indicated that the positive effect on the photosynthetic characteristics of *E. ferox* leaves was most significant when 18.8 kg·667 m^−2^ of SRU was applied at one time at AFP20.

### 2.5. Effects of Different Periods of SRU on the Yield of E. ferox

The effect of SRU treatment at the S1-S5 period on fruit growth, development, and yield of *E. ferox* DAF15-DAF40 was investigated using 18.8 kg·667 m^−2^ as the quantity of application. As the period of SRU application was delayed, the net fruit weight, fruit diameter, number of grains per fruit, grains per fruit, and seed kernel weight increased gradually with the growth and development of the plants, and all of them showed a trend of increasing and then decreasing. The maximum value was reached at S2 treatment, which was significantly higher than that of CK (Figure 6). In addition, there were significant differences in *E. ferox* yield, in which the maximum *E. ferox* yield of S2 treatment reached 8.10 kg, and the yield of a single plant showed a sequence of S2 > S1 > CK > S4 > S3. The total yield in the S2 treatment reached a maximum of 31.03 kg, which showed a sequence of S2 > S4 > CK > S1 > S3 (Appendix A). The above results indicated that at AFP20, a one-time application of 18.8 kg·667 m^−2^ (T3) of SRU had the most significant effect on *E. ferox* growth and development and yield.

### 2.6. Effects of Different Periods of SRU on the Quality of E. ferox

We further investigated the effects of SRU treatments on soluble sugars, starch, and flavonoids in *E. ferox* at different periods. The results of the soluble sugar assay showed that the soluble sugar content in the seed kernel at DAF20 treated at S2 period reached the highest value of 10.30 mg·g^−1^, which increased by 36.4% compared with CK, and the soluble sugar content showed S2 > S4 > S1 > CK > S3 (Figure 7). Further, the seed kernel at DAF20 was selected to measure the relative expression of sugar transporter genes in S2 and CK treatments. The results showed that the relative expression of *EfSWEET11.1*, *EfSWEET11.2*, *EfSWEET11.3*, *EfSWEET4.3*, *EfSWEET1.4*, and *EfSWEET1.6* was significantly higher than that of CK. And are 5.2, 2.5, 3.4, 1.5, 1.8, and 2.5 times higher than CK, respectively (Figure 8). Thus, at AFP20, a single application of 18.8 kg·667 m^−2^ (T3) of SRU had the most significant effect on soluble sugars.

The results of starch content of *E. ferox* seed kernels under different fertilisation periods showed that the total starch in the seed kernels gradually accumulated and was highest at DAF40. The total starch in *E. ferox* seed kernels showed a trend of increasing and then decreasing at 40 days, with the maximum value reached in the S2 treatment, and the difference was significant (Appendix A). The relative expression of starch synthesis genes was further determined in the E. ferox seed kernels of S2 and CK. The relative expression of *EfDBE* was significantly higher than that of CK in DAF20, DAF30, and DAF35, with 0.72, 0.74, and 0.64 times more compared to CK, respectively. The relative expression of *EfSS1*, *EfSBE*, and *EfGBSS* was significantly higher in DAF25-DAF30, DAF25-DAF35, and DAF20-DAF30 than in CK, respectively (Figure 8). Finally, *E. ferox* seed kernels fertilised (S2) were selected for the determination of SSS, SBE, and GBSS enzyme activities. The results showed that SSS, SBE, and GBSS enzyme activities were significantly higher than those of CK at DAF15-DAF40, DAF20-DAF40, and DAF15-DAF40, and reached the maximum value at 30, 25, and 30 days, respectively (Appendix A). In summary, DAF25-DAF30, S2 treatment significantly increased both the expression of *EfSBE*, *EfGBSS*, and *EfSS1* genes and the activities of the corresponding starch synthases SBE, GBSS, and SSS.

Finally, we determined the changes in flavonoid compound content. The data showed a gradual decrease in flavonoid content from DAF15 to DAF40, and the S2 treatment had the highest flavonoid compound content of 3.53 mg·g^−1^, which was 22.80% higher than that of CK (Appendix A). Therefore, the relative expression of flavonoid synthesis genes was determined in *E. ferox* seed kernels from S2 and CK at DAF15. The results showed that the relative expression of *EfPAL1*, *EfPAL2*, *EfF3H3*, *EfDFR*, *EfLAR*, and *EfFLS* was significantly higher than that of CK, with 2.6-fold, 1.6-fold, 1.2-fold, 2.1-fold, 1.1-fold, and 1.0-fold increases, respectively (Figure 8). In summary, a single application of 18.8 kg·667 m^−2^ (T3) SRU at AFP20 was able to provide significant quality of *E. ferox*. Compared with traditional fertilization, we have studied the fertilization pattern more precisely, which is important to know in production.

## 3. Discussion

Plant leaves are the main place of photosynthesis, and chlorophyll in leaves is the main pigment of photosynthesis in plants. Its content can reflect the ability of photosynthesis in plants to a certain extent [24]. Reasonable application of nitrogen fertiliser can increase the chlorophyll content of leaves. The higher the content, the stronger the photosynthesis and the more conducive to the accumulation of carbohydrates in crops, thus increasing crop yields [25]. Nitrogen fertiliser applied in the pre-emergence period increased the net photosynthetic rate of leaves and was conducive to the improvement in apple yield and quality in apple [26]. A one-time application of SRU treatment at the first standing leaf stage of lotus root can increase the number of standing leaves, leaf area, and SPAD value of lotus root and enhance the photosynthetic efficiency of leaves in order to increase yield [27]. However, studies on the enhancement in *E. ferox*’s yield and quality after SRU treatment have been rarely discussed. In the present study, the application of 18.8 kg·667 m^−2^ of SRU at AFP20 significantly increased the chlorophyll content of *E. ferox* leaves. This is likely to enhance the photosynthetic capacity of the leaves, thereby improving the yield and quality of the *E. ferox*. This is a lesser amount and lower cost of slow-release fertilizer compared to previous studies.

During fertiliser application, different application periods and application rates are particularly critical for soluble sugar accumulation and transport. At present, few studies have been reported on the effect of SRU on crop soluble sugars, but the effect of fertiliser application on this has been reported. Compared to other treatments, the application of slow-release fertiliser at different stages of growth and development of sugarcane resulted in significantly higher soluble sugar content during the elongation and flowering stages [28]. Application of urea at the primordial stage (DAF15) resulted in a significantly higher soluble sugar content than the other treatments in cotton [29]. In *E. ferox*, the soluble sugar content was significantly increased by a one-time application of SRU at 18.8 kg·667 m^−2^ at AFP20. This is consistent with the results of previous studies, but there were differences in the period of fertilization. Urea application at 210 kg·ha^−1^ increased soluble sugar content and the expression of relative sugar transporter genes in coral lettuce, providing sufficient polysaccharides for root assimilation [30]. In *E. ferox*, the relative expression of sugar transporter genes (*EfSWEET11.3*, etc.) in the seed kernels was significantly increased. This result is consistent with previous studies, suggesting that the timed and quantitative application of SRU can promote sugar metabolism in the seed kernel of *E. ferox* and ultimately affect the various life activities of the plant.

Slow-release urea affects crop starch content by regulating the expression of key genes for starch synthesis [14]. At present, few studies have been reported on the effect of SRU quantity of application and period of use on starch synthesis in crops, but the effect of nitrogen fertiliser application on this has been reported. With the increase in nitrogen application, the expression of starch synthesis-related genes *GBSSI*, *SBEI*, and *SSIII* in early and mid-growth potatoes showed a tendency to increase and then decrease [31]. Our results showed that the expression of *EfDBE*, *EfSS1*, *EfSBE*, and *EfGBSS* was significantly increased by a one-time application of 18.8 kg·667 m^−2^ SRU at AFP20, which ultimately led to an increase in *E. ferox* yield. Urea application at 240 kg·hm^−2^ increased SSS expression during wheat grain development, which increased SSS enzyme activity, which in turn changed the composition and distribution of starch grains and ultimately improved starch quality [32]. Application of 225 kg∙hm^−2^ urea increased the expression level of *OsGBSS I* at the beginning of filling, which in turn increased its GBSS enzyme activity and total starch content and improved the quality of rice [33]. In *E. ferox*, the corresponding enzyme activities of SBE, GBSS, and SSS increase significantly. Consistent with the results of previous studies.

As a medicinal plant, the content of flavonoid compounds in the seed kernels is an important factor affecting its quality [34]. Slow-release urea applications can affect the synthesis and accumulation of flavonoid compounds in crops [35]. The content of flavonoid compounds in the leaves of *Picea rubra* within a certain range increased with the increase in the amount of slow-release fertiliser, in which the highest content of flavonoid compounds in the leaves of the treatment of slow-release fertiliser paired with urea was increased by 33.06% compared with CK [36]. However, the study of the effect of a slow-release urea application period on the content of flavonoid compounds and the expression of related genes in crops has not been reported. In this study, it was clarified that SRU promoted the expression of *EfPAL*, *EfF3H*, and other genes in *E. ferox* when SRU was applied at one time for AFP20, which in turn increased the content of flavonoid compounds within *E. ferox* seed kernels, providing theoretical support for the application of fertilizers to enhance the quality of *E. ferox* seed kernels.

## 4. Materials and Methods

### 4.1. Plant Materials and Experimental Design

The experiments were conducted in 2022 and 2023 at the aquatic vegetable Experimental Field of Yangzhou University in Jiangsu Province, China. The *E. ferox* variety used in the test is ‘ZHSQ’, and the planting date is 1 May.

Each plant was independently planted in a 12 m^2^ open concrete experimental plot, and all management was consistent and good. Sampling was conducted by selecting floating leaves and taking three leaves of 5cm × 5 cm in a straight line from the center to the edge of the leaf, evenly spaced. For each leaf and fruit, three samples were taken from each of the three equally treated experimental plots for data determination. After sampling, we analyzed the physiological data of each fruit separately (yield in terms of the weight of the whole fruit) and measured the indicators of the edible part (Figure 9).

We determined the range of fertilizers and other variables to be used based on the pre-laboratory research base [10], and then we confirmed the final application rates through pre-experimentation. When applying the fertilizer, we evenly distributed eight points around a 1 m radius of the plant roots and applied the fertilizer evenly to the soil at each point (Figure 9A).

There were six treatments in the experiment, and 17.8 kg·667 m^−2^ urea treatment (CK) was set as the control group. Based on the total nitrogen content of the control group, the total nitrogen content was reduced by 40% treatment (T1 = 11.3 kg·667 m^−2^). The total nitrogen content was reduced by 20% treatment (T2 = 15.0 kg·667 m^−2^). The total nitrogen amount was equal to the slow-release nitrogen fertilizer treatment (T3 = 18.8 kg·667 m^−2^). The total nitrogen amount was increased by 20% treatment (T4 = 22.6 kg·667 m^−2^). Total nitrogen was increased by 40% for treatment (T5 = 26.3 kg·667 m^−2^). Each treatment was repeated three times, all in a single application two weeks after colonization. The amount of phosphorus and potassium fertilizer in each treatment was consistent (K_2_O = 70.2 kg·667 m^−2^ and P_2_O_5_ = 87.6 kg·667 m^−2^).

The experiment (2023) was arranged in an experimental plot with an area of 19 m^2^. The control group (CK) was set up with 17.8 kg·667 m^−2^ ordinary urea. After fixed planting (AFP0), a one-time application of SRU 18.8 kg·667 m^−2^ (S1); A one-time application of SRU 18.8 kg·667 m^−2^ (S2) was applied 20 days after fixed planting (AFP20). A one-time application of SRU 18.8 kg·667 m^−2^ (S3) was applied 30 days after fixed planting (AFP30). A one-time application of SRU 18.8 kg·667 m^−2^ (S4) was applied 15 days after flowering. Urea treatment (CK) 17.8 kg·667 m^−2^ was applied evenly in the third, fifth, and seventh weeks after planting. The application method of superphosphate and potassium sulfate is the same as above.

### 4.2. Sampling Method

During the growth period of *E. ferox*, the sample of *E. ferox* fruit is divided into six times. Fruit samples were taken once after 15 days, 20 days, 25 days, 30 days, 35 days, and 40 days after flowering (DAF15, DAF20, DAF25, DAF30, DAF35, and DAF40, respectively). Three fruits were randomly selected for each treatment for subsequent experiments.

### 4.3. E. ferox Fruit Yield Determination

At the end of the *E. ferox* growth period, all the fruits are collected and weighed to measure the yield. See Appendix A for data.

### 4.4. Determination of SPAD in E. ferox Leaves

SPAD values were measured 25 days after flowering using a portable chlorophyll meter (SPAD-502). Three leaves with consistent growth and good growth were selected from each experimental plot. For each leaf, 15 parts were randomly selected to measure the SPAD value, and the average value was taken as the SPAD value of this treatment. See Appendix A for data.

### 4.5. Determination of Chlorophyll Content in E. ferox Leaves

The chlorophyll content of *E. ferox* leaves was determined by using the chlorophyll content detection kit (Beijing Solarbio Science & Technology Co., Ltd., Beijing, China). Chlorophyll a and chlorophyll b have maximum absorbance at wavelengths of 645 nm versus 663 nm. By applying specific formulae, the content of chlorophyll a, chlorophyll b, and total chlorophyll can be estimated [37]. See Appendix A for data.

### 4.6. Determination of Starch Content in Seed Kernel of E. ferox

A starch content detection kit (Beijing Solarbio Science & Technology Co., Ltd., Beijing, China) was used to determine the total starch content in the seed kernel of *E. ferox* [38]. The calculation for the content of amylopectin was as follows: amylopectin = total starch content—amylose content. Each sample was set up for three biological replicates [39]. See Appendix A for data.

### 4.7. Determination of Soluble Sugar Content in Seed Kernel of E. ferox

A soluble sugar content detection kit (Beijing Solarbio Science & Technology Co., Ltd., Beijing, China) was used to determine the Soluble sugar content in the seed kernel of *E. ferox*. See Appendix A for data.

### 4.8. Determination of Enzyme Activity Related to Starch Synthesis in Seed Kernel of E. ferox

GBSS, SBE, and SSS activity detection kit (Beijing Solarbio Science & Technology Co., Ltd., Beijing, China) were used to determine the activities of GBSS, SBE, and SSS, the key enzymes of starch synthesis. See Appendix A for data.

### 4.9. Determination of Relative Expression Levels of Genes Related to Starch Synthesis and Sugar Transport in Seed Kernel of E. Ferox

The seed kernel RNA of *E. ferox* was extracted using the polysaccharide polyphenol plant total RNA extraction kit (Shanghai Pudi Biotechnology Co., Ltd., Shanghai, China). After extraction, agarose gel electrophoresis and a spectrophotometer were used to detect the RNA content and quality. First-strand cDNAs were obtained using the reverse transcription kit provided by Nanjing Nuoweizen Biological Company, Nanjing, China. Reverse transcription is required to be performed at 25 °C, 10 min; 50 °C, 15 min; 85 °C, 5 min. The fluorescence quantitative primers of related genes were designed using the NCBI website, and their specificity was guaranteed with the genome database blastp. Fluorescence quantitative PCR was performed according to the kit of Shanghai Novizan Biotechnology Co., LTD, Shanghai, China. See Appendix A for data.

qRT-PCR system: cDNA 2 μL, 7.2 μL SYBR Mix, F/R primers 0.4 μL, 10 μL ddH_2_O. The reaction system of qRT-PCR was predenaturated at 95 °C for 5 min, denatured at 10 s at 95 °C, and extended for 30 s at 60 °C for a total of 40 cycles. The relative expression △data of genes were collated and calculated using the 2^−∆∆Ct^ method. Specific primers for the synthesis of target genes and internal reference genes are in Appendix A.

### 4.10. Data Processing and Analysis

Microsoft Excel 2018 software was used for data entry, organizing, and tabulation of experimental data. Bar charts and line graphs were made using GraphPad 8.0 software. Data were analyzed by one-way ANOVA using SPSS 26.0 statistical software.

## 5. Conclusions

In summary, the reasons for increasing the yield of *E. ferox* in this study were to improve the photosynthetic capacity of *E. ferox* leaves by increasing the content of leaf chlorophyll, increasing the expression of sugar transporter genes, thereby promoting the operation of soluble sugars, and provide sufficient substrates for starch synthesis in seed kernels. Then, by promoting the activity of enzymes related to starch synthesis, starch accumulation in the seed kernel is promoted, which ultimately increases the yield of *E. ferox*. In addition, the application of SRU increased the content of flavonoids, soluble sugars, and starch in the seed kernels, thus improving the quality of *E. ferox*. Therefore, this study mainly focused on the physiological, yield, and quality aspects of *E. ferox*, and there are limitations in molecular studies. However, the metabolic processes of substances in plants are interrelated, and the effects of SRU on crops are holistic. Therefore, aspects such as molecular biology deserve further in-depth studies to elucidate the molecular regulatory mechanisms behind them.

## Figures and Tables

**Figure 1 ijms-25-11737-f001:**
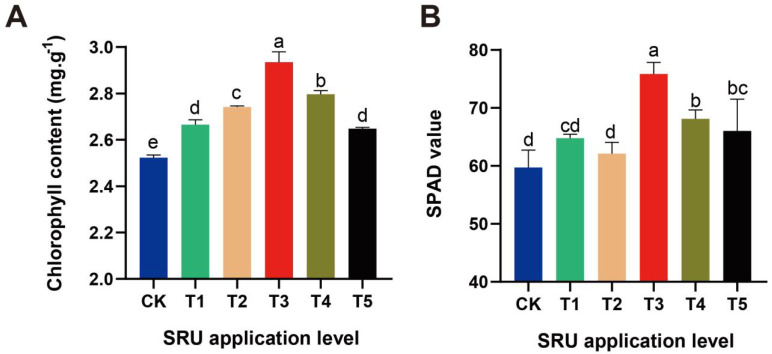
The photosynthesis-related physiological indexes of *E. ferox* leaves under different SRU quantities of application (CK and T1−T5). (**A**) Chlorophyll content. (**B**) SPAD value. Different letters indicate significantly different values (*p* < 0.05).

**Figure 2 ijms-25-11737-f002:**
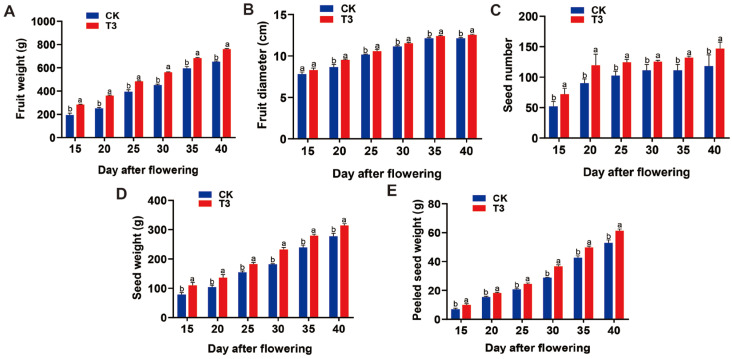
Physiological data of *E. ferox* under different SRU quantities of application (CK and T3). (**A**) Fruit weight. (**B**) Fruit diameter. (**C**) Seed number. (**D**) Seed weight. (**E**) Peeled seed weight. Different letters indicate significantly different values (*p* < 0.05).

**Figure 3 ijms-25-11737-f003:**
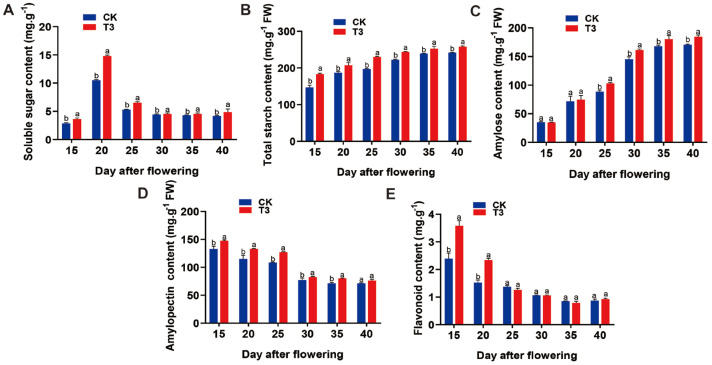
Content of each indicator under different SRU quantities of application (CK and T3). (**A**) Soluble sugars. (**B**) Total starch. (**C**) Amylose. (**D**) Amylopectin. (**E**) Flavonoids. Different letters indicate significantly different values (*p* < 0.05).

**Figure 4 ijms-25-11737-f004:**
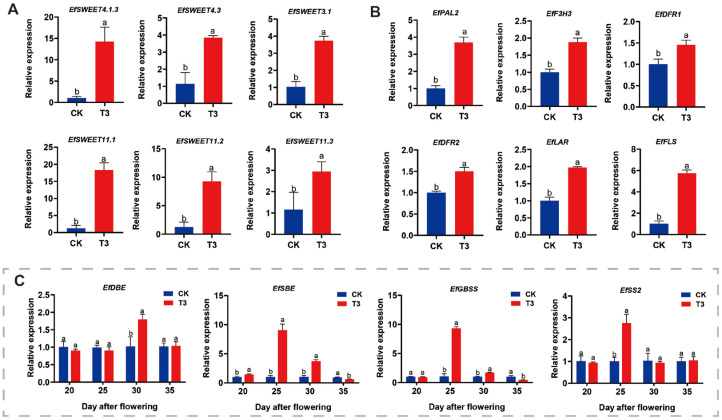
Related gene expression under different SRU quantities of application (CK and T3). (**A**) Soluble sugar-related gene expression. (**B**) Flavonoids synthesis-related gene expression. (**C**) *EfDBE*, *EfSBE*, *EfGBSS*, *EfSS2*. Different letters indicate significantly different values (*p* < 0.05).

**Figure 5 ijms-25-11737-f005:**
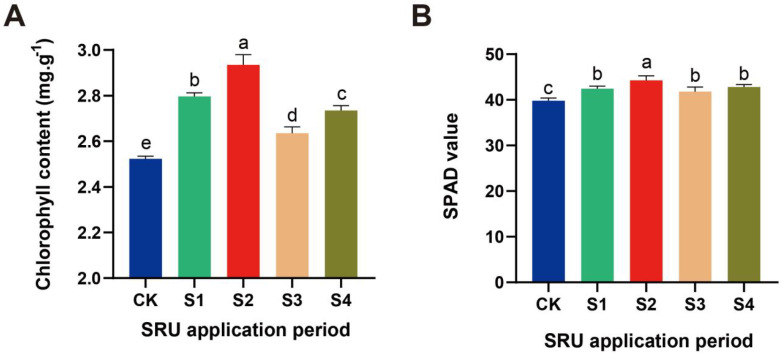
The photosynthesis-related physiological indexes of *E. ferox* leaves under different periods of application of SRU (CK and S1−S4). (**A**) Chlorophyll content. (**B**) SPAD value. Different letters indicate significantly different values (*p* < 0.05).

**Figure 6 ijms-25-11737-f006:**
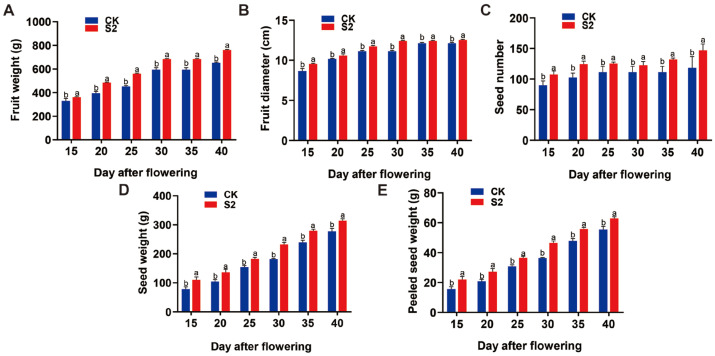
Physiological data of *E. ferox* under different periods of application of SRU (CK and S2). (**A**) Fruit weight. (**B**) Fruit diameter. (**C**) Seed number. (**D**) Seed weight. (**E**) Peeled seed weight. Different letters indicate significantly different values (*p* < 0.05).

**Figure 7 ijms-25-11737-f007:**
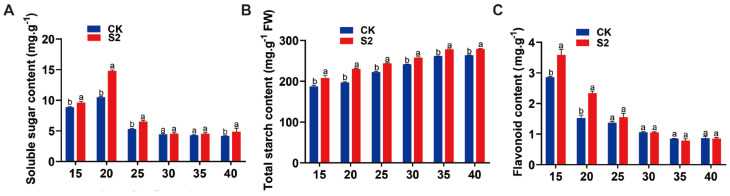
Content of each indicator under different periods of application of SRU (CK and S2). (**A**) Soluble sugars. (**B**) Total starch. (**C**) Flavonoids. Different letters indicate significantly different values (*p* < 0.05).

**Figure 8 ijms-25-11737-f008:**
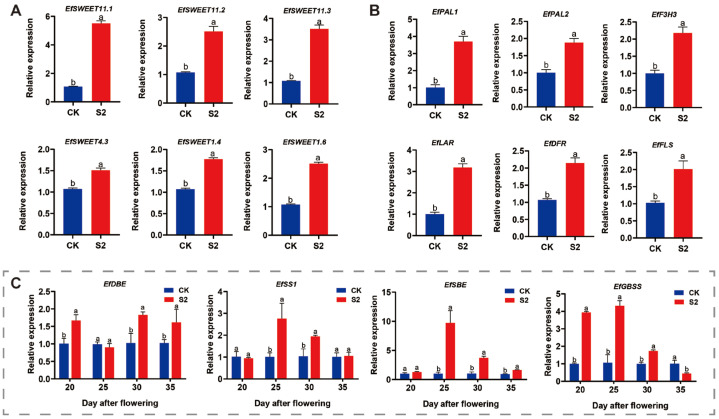
Related gene expression under different periods of application of SRU (CK and S2). (**A**) Soluble sugar-related gene expression. (**B**) Flavonoids synthesis-related gene expression. (**C**) *EfDBE*, *EfSS1*, *EfSBE*, *EfGBSS*. Different letters indicate significantly different values (*p* < 0.05).

**Figure 9 ijms-25-11737-f009:**
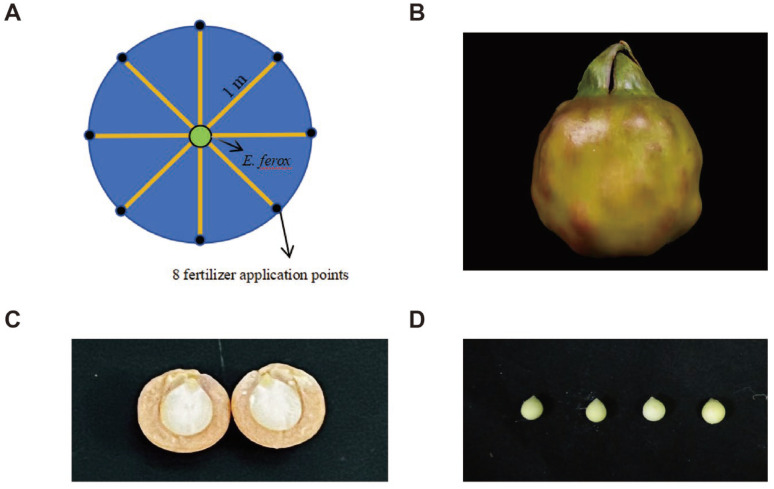
**(A**) Schematic diagram of fertilization of *E. ferox*. (**B**) Diagram of the fruits of *E. ferox*. (**C**) Profile of *E. ferox* seed kernels. (**D**) The edible part of the seed kernels of *E. ferox*.

## Data Availability

The original contributions presented in the study are included in the article. Further inquiries can be directed at the corresponding authors.

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
