# Peer review of "Effect of Slow-Release Urea on Yield and Quality of Euryale ferox"

_ijms, 2024, doi:10.3390/ijms252111737_

Round 1

Reviewer 1 Report

Comments and Suggestions for Authors

The manuscript examines the effects of slow-release urea (SRU) on the yield and quality of Euryale ferox (E. ferox), focusing on various concentrations and application periods. The study provides valuable insights into the influence of SRU on key parameters such as yield, soluble sugar, starch, and flavonoid content. Furthermore, it explores the underlying molecular mechanisms by analyzing the expression of sugar transporter and flavonoid synthesis-related genes. The research is of potential significance for sustainable agriculture, especially for enhancing the nitrogen use efficiency and optimizing fertilizer application for E. ferox.

Minor Comments:

  1. Line 31: Define "SRU" (Slow-Release Urea) upon first use to ensure clarity for the readers.
  2. Line 36-38: Gene names such as EfSWEET and EfPAL should be italicized to follow scientific nomenclature standards.
  3. Line 39: Correct the phrasing of "in depth the effect" to "the in-depth effect."
  4. Line 46: The sentence should be restructured for clarity to:
    "However, most crops have very low nitrogen use efficiency, and more than half of the nitrogen fertilizer applied is lost before it is absorbed by the crop."
  5. Line 66: Clarify the term "transport pairs." If "transporters" was intended, please correct the terminology; otherwise, explain what "transport pairs" refers to.
  6. Lines 82-84: The sentence should be revised for clarity and grammatical accuracy.
  7. Line 85: Add a comma after "nutrient absorption of E. ferox."
  8. Line 131-132: Modify the phrasing to "the relative expression of sugar transporter genes."
  9. Line 214: Italicize E. ferox for consistency.
  10. Line 312: Italicize Picea rubra.
  11. Results Section Titles: For improved clarity, consider revising the section titles in the results. For example:
    • "Effects of Different Quantities of SRU on the Chlorophyll Content of E. ferox"
    • "Effects of Different Quantities of SRU on the Yield of E. ferox"
    • “Effects of Different Quantities of SRU on the Quality of E. ferox”

Major Comments:

  1. Introduction:
    It would be beneficial for the authors to include a paragraph explaining the importance of improving the yield and quality of E. ferox seed kernels. This would provide readers with context and highlight the practical significance of this research.
  2. Limitations and Future Directions:
    The conclusions or discussion section should address the limitations of this study and propose future directions for research. Understanding potential challenges and areas for further exploration can strengthen the impact of the findings.
  3. Cost Implications:
    The study would benefit from an economic analysis comparing the cost-effectiveness of slow-release urea versus conventional urea for farmers. This is an important consideration for practical implementation.
  4. Urea/Nitrogen Residuals:
    It is important to include data on residual nitrogen or urea levels in the water after harvest in both control (CK) and SRU treatments. This information could provide insights into environmental impact and efficiency of nitrogen utilization.
  5. Algal Bloom/Eutrophication:
    Given that excess nitrogen can lead to eutrophication, it would be valuable to discuss whether the use of SRU reduced the occurrence of algal blooms compared to conventional fertilizers. This is particularly relevant in aquatic field systems where nutrient runoff could lead to environmental damage.

Overall Recommendation:

The manuscript presents an important and timely study on the use of slow-release urea in E. ferox. While the research is sound and valuable, it could be strengthened by addressing the major points listed above, particularly regarding environmental impact, economic implications, and the potential limitations of the study. Addressing these issues will make the manuscript more comprehensive and practical for broader application.

Comments on the Quality of English Language

In terms of the quality of the English language, the manuscript requires some improvement for clarity, grammar, and scientific precision. While the overall content is understandable, there are several instances where the phrasing is awkward, and some sentences are either incomplete or lack proper structure. Additionally, certain scientific terms and gene names should adhere to the appropriate conventions (e.g., italicization).

Author Response

Minor Comments:

Comment 1: Line 31: Define "SRU" (Slow-Release Urea) upon first use to ensure clarity for the readers.

Response 1: Thanks for your comments. This section has been revised. (See Page 1, Lines 31)

Comment 2: Line 36-38: Gene names such as EfSWEET and EfPAL should be italicized to follow scientific nomenclature standards.

Response 2: Thanks for your comments. This section has been revised.(See Page 1, Lines 37)

Comment 3: Line 39: Correct the phrasing of "in depth the effect" to "the in-depth effect."

Response 3: Thanks for your comments.This section has been revised. (See Page 1, Lines 40)

Comment 4: Line 46: The sentence should be restructured for clarity to:
"However, most crops have very low nitrogen use efficiency, and more than half of the nitrogen fertilizer applied is lost before it is absorbed by the crop."

Response 4: Thanks for your comments. This section has been revised. (See Page 1, Lines 49-50)

Comment 5: Line 66: Clarify the term "transport pairs." If "transporters" was intended, please correct the terminology; otherwise, explain what "transport pairs" refers to.

Response 5: Thanks for your comments. This section has been revised.(See Page 2, Lines 70)

Comment 6:Lines 82-84: The sentence should be revised for clarity and grammatical accuracy.

Response 6: Thanks for your comments. This section has been revised. As follows: The expression levels of starch-related genes OsSBEII, OsGBSSI, and OsSBEI were significantly decreased by SRU treatment, which reduced the ratio of straight-chain starch content to branched-chain starch content and improved the cooking quality of rice.(See Page 2, Lines 83-86)

Comment 7: Line 85: Add a comma after "nutrient absorption of E. ferox."

Response 7: Thanks for your comments. This section has been revised. (See Page 2, Lines 98)

Comment 8: Line 131-132: Modify the phrasing to "the relative expression of sugar transporter genes."

Response 8: Thanks for your comments. This section has been revised. (See Page 4, Lines 146)

Comment 9:Line 214: Italicize E. ferox for consistency.

Response 9: Thanks for your comments. This section has been revised. (See Page 7, Lines 230)

Comment 10:Line 312: Italicize Picea rubra.

Response 10: Thanks for your comments. This section has been revised. (See Page 9, Lines 336)

Comment 11:Results Section Titles: For improved clarity, consider revising the section titles in the results. For example:

"Effects of Different Quantities of SRU on the Chlorophyll Content of E. ferox"

"Effects of Different Quantities of SRU on the Yield of E. ferox"

“Effects of Different Quantities of SRU on the Quality of E. ferox”

Response 11: Thanks for your comments. This section has been revised. (See Page 3-7)

Major Comments:

Comment 1: Introduction: It would be beneficial for the authors to include a paragraph explaining the importance of improving the yield and quality of E. ferox seed kernels. This would provide readers with context and highlight the practical significance of this research.

Response 1: Thanks for your comments. This section has been revised. As follows: With the increasing demand for products, improving yield and quality has become an important issue.(See Page 1, Lines 46-47)

Comment 2:Limitations and Future Directions: The conclusions or discussion section should address the limitations of this study and propose future directions for research. Understanding potential challenges and areas for further exploration can strengthen the impact of the findings.

Response 2: Thanks for your comments. This section has been revised. As follows: In summary, the reasons for increasing the yield of E. ferox in this study were to improve the photosynthetic capacity of E. ferox leaves by increasing the content of leaf chlorophyll, increase the expression of sugar transporter genes thereby promoting the operation of soluble sugars, and provide sufficient substrates for starch synthesis in seed kernels. Then by promoting the activity of enzymes related to starch synthesis, starch accumulation in the seed kernel is promoted, which ultimately increases the yield of E. ferox. In addition, the application of SRU increased the content of flavonoids, soluble sugars and starch in the seed kernels, thus improving the quality of E. ferox. Therefore, this study mainly focused on the physiological, yield, and quality aspects of E. ferox, and there are limitations in molecular studies. However, the metabolic processes of substances in plants are interrelated and the effects of SRU on crops are holistic, so aspects such as molecular biology deserve further in-depth studies to elucidate the molecular regulatory mechanisms behind them.(See Page 9, Lines 432-444)

Comment 3: Cost Implications: The study would benefit from an economic analysis comparing the cost-effectiveness of slow-release urea versus conventional urea for farmers. This is an important consideration for practical implementation.

Response 3: Thanks for your comments. The cost of fertilizer application includes the cost of fertilizer purchase and labor cost. SRU is a one-time fertilizer application, the number of times of fertilizer application is significantly less than the traditional fertilizer application, which can save the amount of fertilizer and labor cost. And the high utilization rate of SRU will also reduce the fertilizer usage. At the same time, the crop yield is significantly higher than that of traditional fertilization, and more considerable economic benefits are obtained(Yu et al., 2022).

Comment 4: Urea/Nitrogen Residuals: It is important to include data on residual nitrogen or urea levels in the water after harvest in both control (CK) and SRU treatments. This information could provide insights into environmental impact and efficiency of nitrogen utilization.

Response 4: Thanks for your comments. Determination of fertilizer residues has limitations for the following reasons:

1) E. ferox is an aquatic plant, high temperature in summer will lead to faster evaporation of water, and need to add water in time, which will lead to the level of urea in the water is greatly affected.

2) Fertilizer is applied to the soil, its main should be in the soil by the roots dissolved and absorbed, levels in water are relatively small and unstable. In summary, whether in soil or water, current measurements are not very responsive to residues, and further research is needed to develop better methods to solve to this issue.

Comment 5: Algal Bloom/Eutrophication: Given that excess nitrogen can lead to eutrophication, it would be valuable to discuss whether the use of SRU reduced the occurrence of algal blooms compared to conventional fertilizers. This is particularly relevant in aquatic field systems where nutrient runoff could lead to environmental damage.

Response 5: Thanks for your comments. First of all, ordinary fertilizers are applied many times and in large quantities, can lead to eutrophication of water bodies. SRU is a one-time application of fertilizer that is released slowly throughout the life cycle, which does not result in a short-term overdose of fertilizer and thus reduces the probability of eutrophication of water bodies. Secondly, the slow release of SRU is in line with the law of plant growth and development, and the utilization rate is high, which will not lead to eutrophication of water bodies(Wu et al., 2017).

Reviewer 2 Report

Comments and Suggestions for Authors

The peer-reviewed manuscript „Effect of Slow-release Urea on Yield and Quality of Euryale ferox" by Peng Wua and Co-authors, concerns the possibility of increasing the yield and at the same time increasing the starch content in seeds of Euryale ferox. Euryale ferox is a very interesting freshwater plant from southern and eastern Asia belonging to the water lily family Nymphaeaceae. Certainly is worth the attention of researchers. The issues discussed in the presented work are important for its commercial breeding and are interesting. However, in my opinion, the work requires improvement.

Comments

It would be good not to repeat words that are in the title, increasing the possibility of finding publications in databases

In my opinion, it would be good to include information about the research object in the introduction: what Euryale ferox is, what is its ecology, why is it cultivated, what is its economic importance, whether only its seeds or also the fruit pulp is used (also in two sentences how the fruit and seeds are constructed).

There is no information for what reason this plant is an important object of research, and first of all, what is the aim of the research, and what are the research hypotheses.

The results are presented quite illegibly. It would be worth rethinking the form of the graphs, the font used on the graphs is too small. In my opinion, it would be worth explaining some abbreviations, it is not obvious to everyone, Fragments interpreting the results should rather to move on the discussion. In general, It would be beneficial for understanding the work and to interest readers in this issue to show the results more transparent way.

The discussion, unfortunately, is rather a review of the literature. There is a lack of confrontation of one's results with the literature, and showing their significance.  The proposed approach has a very practical dimension, the results are presented rather in agricultural terms. Perhaps it would be worth trying to interpret them biologically.

In my opinion, the methodological part also requires expansion and supplementation. In the case of the methods used, their precise description is of course not necessary, but it would be good to provide an appropriate citation.

Many things need to be clarified, e.g. whether the research environment was homogeneous (e.g. whether the light conditions were the same in hole the experiment), whether the experiment was conducted in an open area or under controlled (partially controlled) conditions, what leaves were selected for analysis (only floating or also submerged), what region of leaf blades were used (from the edge of the leaf, from its center?), how large a leaf surface was sampled, whether mechanical damage to the leaf during samples could have had any impact on the further development of the plant, what was the design of the experiment, whether blocks of repetitions were used, whether there were only biological repetitions, whether each fruit was analyzed independently, whether the sample was combined, whether whole fruit, the fruit pulp or only the seeds (the whole seed or an isolated part?) were analyzed, what statistical methods were used to assess significance, what the error bars on the graphs mean, on what basis was the control dose of fertilizer determined, on what basis were the other experimental variants determined, how the fertilizer was applied, whether the plants exposed to different doses were isolated from each other (or were they grown in separate tanks).

I would also like to know how Euryale ferox is cultivated, whether these are specially designated artificial reservoirs or natural lakes and ponds are being exploited, how the fertilizers used can affect the entire ecosystem of the lake, its flora and fauna, whether the proposed dose of fertilizer can affect the eutrophication of the reservoir, what could be the consequences of this? What is the scale of cultivation/production of fruit? Are these plants commonly cultivated or rather naturally growing plants used? What is the commercial importance of Euryale ferox? What is this plant used for, whether it is important for the industry, and in which type of industries it can be used. Unfortunately, this plant is not commonly known in Europe but after reading this text this is interesting for me.

In conclusion, I think the manuscript could be suitable for publication in the journal, but after major corrections.

Author Response

Comment 1: It would be good not to repeat words that are in the title, increasing the possibility of finding publications in databases.

Response 1: Thanks for your comments. We have optimized.

Comment 2: In my opinion, it would be good to include information about the research object in the introduction: what Euryale ferox is, what is its ecology, why is it cultivated, what is its economic importance, whether only its seeds or also the fruit pulp is used (also in two sentences how the fruit and seeds are constructed).

Response 2: Thanks for your comments. This section has been revised. As follows: 

Euryale ferox Salisb. is an annual macrophyte aquatic herb that belongs to the genus Euryale from family Nymphaeaceae(Wu et al., 2021). Euryale ferox has a long history of cultivation in China, is a kind of medicine and food plants(Wu et al., 2022). Not only has a high nutritional value, but also has to tonify the middle and benefit the qi, beneficial to the kidneys to consolidate the essence, to remove the dampness of the spleen and other health care effects, is the source of many Chinese herbal medicine(Song et al., 2010). At the same time, Euryale ferox in the domestic and international market demand is rapidly increasing, the price is rising, the development prospects are very broad.(See Page 2, Lines 90-97)

Comment:There is no information for what reason this plant is an important object of research, and first of all, what is the aim of the research, and what are the research hypotheses.

Response 3: Thanks for your comments. Point2 has already explained its importance and the purpose has been made clear.

Comment 4:The results are presented quite illegibly. It would be worth rethinking the form of the graphs, the font used on the graphs is too small. In my opinion, it would be worth explaining some abbreviations, it is not obvious to everyone, Fragments interpreting the results should rather to move on the discussion. In general, It would be beneficial for understanding the work and to interest readers in this issue to show the results more transparent way.

Response 4: Thanks for your comments. Images have been optimized. All abbreviations have been described in Materials and Methods. Since the results section drives the next conclusion by summarizing several small conclusions, we have added a staged discussion. As follows: 

1) In summary, the optimal quantity of application of SRU was 18.8 kg·667 m-2. Apply less fertilizer and lower costs than traditional fertilization.(See Page 5, Lines 184-185).

2) Compared with the traditional fertilization, we have studied the fertilization pattern more precisely, which is important to know in production.(See Page 8, Lines 271-272)

Comment 5:The discussion, unfortunately, is rather a review of the literature. There is a lack of confrontation of one's results with the literature, and showing their significance.  The proposed approach has a very practical dimension, the results are presented rather in agricultural terms. Perhaps it would be worth trying to interpret them biologically.

Response 5: Thanks for your comments. The discussion section has been revised. As follows: 

1) Application of urea at the primordial stage (DAF15) resulted in a significantly higher soluble sugar content than the other treatments in cotton(Zhang et al., 2014). In E. ferox, the soluble sugar content were significantly increased by a one-time application of SRU at 18.8 kg·667 m-2 at AFP20. This is consistent with the results of previous studies, but there were differences in the period of fertilization. Urea application at 210 kg·ha-1 increased soluble sugar content and the expression of relative sugar transporter genes in coral lettuce, providing sufficient polysaccharides for root assimilation(Hou et al., 2007). In E. ferox, the relative expression of sugar transporter genes (EfSWEET11.3, etc.) in the seed kernels were significantly increased. This result is consistent with previous studies. Which suggests that the timed and quantitative application of SRU can promote sugar metabolism in the seed kernel of E. ferox, and ultimately affect the various life activities of the plant. (See Page 9, Lines 305-316).

2) Our results showed that the expression of EfDBE, EfSS1, EfSBE, EfGBSS were significantly increased by a one-time application of 18.8 kg·667 m-2 SRU at AFP20, which ultimately led to an increase in E. ferox yield. Urea application at 240 kg·hm-2 increased SSS expression during wheat grain development, which increased SSS enzyme activity, which in turn changed the composition and distribution of starch grains and ultimately improved starch quality (Ran et al., 2020). Application of 225 kg∙hm-2 urea increased the expression level of OsGBSS I at the beginning of filling, which in turn increased its GBSS enzyme activity and total starch content, and improved the quality of rice(Sun et al., 2018). In E. ferox, the corresponding enzyme activities of SBE, GBSS, and SSS increase significantly. Consistent with the results of previous studies.(See Page 9, Lines 323-332).

3)  In summary, the reasons for increasing the yield of E. ferox in this study were to improve the photosynthetic capacity of E. ferox leaves by increasing the content of leaf chlorophyll, increase the expression of sugar transporter genes thereby promoting the operation of soluble sugars, and provide sufficient substrates for starch synthesis in seed kernels. Then by promoting the activity of enzymes related to starch synthesis, starch accumulation in the seed kernel is promoted, which ultimately increases the yield of E. ferox. In addition, the application of SRU increased the content of flavonoids, soluble sugars and starch in the seed kernels, thus improving the quality of E. ferox. Therefore, this study mainly focused on the physiological, yield, and quality aspects of E. ferox, and there are limitations in molecular studies. However, the metabolic processes of substances in plants are interrelated and the effects of SRU on crops are holistic, so aspects such as molecular biology deserve further in-depth studies to elucidate the molecular regulatory mechanisms behind them.(See Page 9, Lines 432-444).

Comment 6:In my opinion, the methodological part also requires expansion and supplementation. In the case of the methods used, their precise description is of course not necessary, but it would be good to provide an appropriate citation.

Response 6: Thanks for your comments. We have optimized. The use of literature in Materials and Methods has been increased.(See Page 10, Lines 399; Page 11, Lines 402-404)

Comment 7:Many things need to be clarified, e.g. whether the research environment was homogeneous (e.g. whether the light conditions were the same in hole the experiment), whether the experiment was conducted in an open area or under controlled (partially controlled) conditions, what leaves were selected for analysis (only floating or also submerged), what region of leaf blades were used (from the edge of the leaf, from its center?), how large a leaf surface was sampled, whether mechanical damage to the leaf during samples could have had any impact on the further development of the plant, what was the design of the experiment, whether blocks of repetitions were used, whether there were only biological repetitions, whether each fruit was analyzed independently, whether the sample was combined, whether whole fruit, the fruit pulp or only the seeds (the whole seed or an isolated part?) were analyzed, what statistical methods were used to assess significance, what the error bars on the graphs mean, on what basis was the control dose of fertilizer determined, on what basis were the other experimental variants determined, how the fertilizer was applied, whether the plants exposed to different doses were isolated from each other (or were they grown in separate tanks).

Response 7: Thanks for your comments. The experimental field is located at the Aquatic Vegetable Experimental Base of Yangzhou University, Yangzhou City, Jiangsu Province, China (119°E,32°N). Each plant was independently planted in a 3 m × 4 m open concrete experimental plots, and all management was consistent and good. Sampling was done by selecting floating leaves and taking three leaves of 5 cm x 5 cm in a straight line from the center to the edge of the leaf, evenly spaced. For each leaf and fruit, three samples were taken from each of the three equally treated experimental plots for data determination. Because the diameter of the leaves are more than 2 m, our sampling leaves are very small, will not cause mechanical damage and will not affect the normal growth of the plant. After sampling, we analyzed the physiological data of each fruit independently (yield was measured as the weight of the whole fruit) and measured the edible portion of the seed kernel for each index. In addition, analysis of variance software was utilized to assess significance and significance was indicated by error bands on the graphs. We determined the range of fertilizers and other variables to be used based on the pre-laboratory research base (Zhao et al., 2022), and then we confirmed the final application rates through pre-experimentation. When applying the fertilizer, we evenly distributed 8 points around a 1 m radius of the plant roots and applied the fertilizer evenly to the soil at each point (see figure). The above has been added in Materials and Methods.(See Page 10, Lines 368-377)

Fertilizer SchematicFruitsseed kernel profileEdible part

Comment 8: I would also like to know how Euryale ferox is cultivated, whether these are specially designated artificial reservoirs or natural lakes and ponds are being exploited, how the fertilizers used can affect the entire ecosystem of the lake, its flora and fauna, whether the proposed dose of fertilizer can affect the eutrophication of the reservoir, what could be the consequences of this? What is the scale of cultivation/production of fruit? Are these plants commonly cultivated or rather naturally growing plants used? What is the commercial importance of Euryale ferox? What is this plant used for, whether it is important for the industry, and in which type of industries it can be used. Unfortunately, this plant is not commonly known in Europe but after reading this text this is interesting for me.

Response 8: Thanks for your comments. Euryale ferox is grown on about 150,000 acres in Chinese and is divided into cultivated (growing in ponds) and wild (growing in lakes)(Xu et al., 2019). Fertilization of Euryale ferox is generally done in separate paddy field that does not affect the entire lake ecosystem. For eutrophication, first of all, ordinary fertilizers are applied many times and in large quantities, can lead to eutrophication of water bodies. SRU is a one-time application of fertilizer that is released slowly throughout the life cycle, which does not result in a short-term overdose of fertilizer and thus reduces the probability of eutrophication of water bodies. Secondly, the slow release of SRU is in line with the law of plant growth and development, and the utilization rate is high, which will not lead to eutrophication of water bodies(Wu et al., 2017). Euryale ferox Salisb. is an annual macrophyte aquatic herb that belongs to the genus Euryale from family Nymphaeaceae(Wu et al., 2021). Euryale ferox has a long history of cultivation in China, is a kind of medicine and food plants(Wu et al., 2022). Not only has a high nutritional value, but also has to tonify the middle and benefit the qi, beneficial to the kidneys to consolidate the essence, to remove the dampness of the spleen and other health care effects, is the source of many Chinese herbal medicine(Song et al., 2010). At the same time, Euryale ferox in the domestic and international market demand is rapidly increasing, the price is rising, the development prospects are very broad.

Round 2

Reviewer 2 Report

Comments and Suggestions for Authors

Dear Authors,

Thank you for your comprehensive answers and realization into account the comments when improving the manuscript.
I suggest considering as keywords e.g. SPAD value, chlorophyll content, starch content, enzymes of starch synthesis, solube sugars,  and instead of E. ferox using its common names e.g. prickly waterlily or Gorgon plant.
Still lack a point "statistical analyses" in the methods (such analyses were performed, it is necessary to provide  the type of statistical tests, software, etc.)
In the manuscript would be worth giving the experimental scheme and photos from "the Authors' Responses".
In the introduction it is worth giving the entire point 8 of "the responses", maybe with a small change - instead of: "Not only has a high nutritional value, but also has to tonify the middle and benefit the qi, beneficial to the kidneys to consolidate the essence, to remove the dampness of the spleen and other health care effects, is the source of many Chinese herbal medicine (Song et al., 2010)" I propose a slightly more general sentence, e.g. "E. ferox has high nutrional value. It is also used in Traditional Chinese Medicine based on the concept of qi (the "vital energy" or 'life energy" of a body), as the component of many Chinese herbal medicine (Song et al., 2010)".
Graphs 3,4,6,7,8 are unfortunately still difficult to read, maybe they can be enlarged, and it is definitely worth increasing the font on them.

Author Response

Comments 1: I suggest considering as keywords e.g. SPAD value, chlorophyll content, starch content, enzymes of starch synthesis, solube sugars,  and instead of E. ferox using its common names e.g. prickly waterlily or Gorgon plant.

Response 1: Thanks for your comments. Prickly waterlily is not just E. ferox, it could be Royal Waterlily. Gorgon plant lacks literature support. E. ferox is more precise and specific and is supported by several papers(Wu et al., 2022ï¼›Jiang et al., 2023).

This section has been revised. As follows: 

Keywords: E. ferox, SRU, SPAD value, Chlorophyll content, Starch content, Solube sugars, Enzymes of starch synthesis, Flavonoids.(See Page 1, Lines 43-44)

Comments 2: Still lack a point "statistical analyses" in the methods (such analyses were performed, it is necessary to provide  the type of statistical tests, software, etc.)

Response 2: Thanks for your comments. This section has been revised. As follows: 

Microsoft Excel 2018 software was used for data entry, organizing and tabulation of the experimental data. Bar charts and line graphs were made using Graphpad 8.0 software. Data were analyzed by one-way ANOVA using SPSS statistical software.(See Page 11, Lines 436-439)

Comments 3: In the manuscript would be worth giving the experimental scheme and photos from "the Authors' Responses".

Response 3: Thanks for your comments. This section has been revised. As follows:

Each plant was independently planted in a 12 m2 open concrete experimental plots, and all management was consistent and good. Sampling was done by selecting floating leaves and taking three leaves of 5cm x 5cm in a straight line from the center to the edge of the leaf, evenly spaced. For each leaf and fruit, three samples were taken from each of the three equally treated experimental plots for data determination. After sampling, we analyzed the physiological data of each fruit separately (yield in terms of the weight of the whole fruit) and measured the indicators of the edible part(Fig. 9).

We determined the range of fertilizers and other variables to be used based on the pre-laboratory research base (Zhao et al., 2022), and then we confirmed the final application rates through pre-experimentation. When applying the fertilizer, we evenly distributed 8 points around a 1 m radius of the plant roots and applied the fertilizer evenly to the soil at each point (Fig. 9A).(See Page 11, Lines 351-362)

Comments 4: In the introduction it is worth giving the entire point 8 of "the responses", maybe with a small change - instead of: "Not only has a high nutritional value, but also has to tonify the middle and benefit the qi, beneficial to the kidneys to consolidate the essence, to remove the dampness of the spleen and other health care effects, is the source of many Chinese herbal medicine (Song et al., 2010)" I propose a slightly more general sentence, e.g. "E. ferox has high nutrional value. It is also used in Traditional Chinese Medicine based on the concept of qi (the "vital energy" or 'life energy" of a body), as the component of many Chinese herbal medicine (Song et al., 2010)".

Response 4: Thanks for your comments. This section has been revised. As follows:

Euryale ferox Salisb. is an annual macrophyte aquatic herb that belongs to the genus Euryale from family Nymphaeaceae(Wu et al., 2021). Euryale ferox has a long history of cultivation in China, is a kind of medicine and food plants(Wu et al., 2022). E. ferox has high nutrional value. It is also used in Traditional Chinese Medicine based on the concept of qi (the "vital energy" or 'life energy" of a body), as the component of many Chinese herbal medicine(Song et al., 2010). At the same time, Euryale ferox in the domestic and international market demand is rapidly increasing, the price is rising, the development prospects are very broad.(See Page 2, Lines 91-98)

Comments 5: Graphs 3,4,6,7,8 are unfortunately still difficult to read, maybe they can be enlarged, and it is definitely worth increasing the font on them.

Response 5: Thanks for your comments. We have optimized.
